# Symmetry control of nanorod superlattice driven by a governing force

Yujia Liang [1,2], Yong Xie[1,3,4], Dongxue Chen[1], Chuanfei Guo[5], Shuai Hou[6], Tao Wen[6], Fengyou Yang[1], Ke Deng[6], Xiaochun Wu[6], Ivan I. Smalyukh [3] & Qian Liu[1,7]

Nanoparticle self-assembly promises scalable fabrication of composite materials with unique properties, but symmetry control of assembled structures remains a challenge. By introducing a governing force in the assembly process, we develop a strategy to control assembly symmetry. As a demonstration, we realize the tetragonal superlattice of octagonal gold nanorods, breaking through the only hexagonal symmetry of the superlattice so far. Surprisingly, such sparse tetragonal superstructure exhibits much higher thermostability than its close-packed hexagonal counterpart. Multiscale modeling reveals that the governing force arises from hierarchical molecular and colloidal interactions. This force dominates the interactions involved in the assembly process and determines the superlattice symmetry, leading to the tetragonal superlattice that becomes energetically favorable over its hexagonal counterpart. This strategy might be instructive for designing assembly of various nanoparticles and may open up a new avenue for realizing diverse assembly structures with pre-engineered properties.

[1] Chinese Academy of Sciences (CAS) Key Laboratory of Nanosystem and Hierarchical Fabrication, CAS Center for Excellence in Nanoscience, National Center for Nanoscience and Technology, Beijing 100190, China. [2] Department of Chemical and Biomolecular Engineering, University of Maryland, Maryland 20742, USA. [3] Department of Physics and Soft Materials Research Center, University of Colorado, Boulder, CO 80309, USA. [4] Department of Physics, Beihang University, Beijing 100191, China. [5] Department of Materials, Southern University of Science and Technology, Shenzhen 518055, China. [6] CAS Key Laboratory of Standardization and Measurement for Nanotechnology, CAS Center for Excellence in Nanoscience National Center for Nanoscience and Technology, Beijing 100190, China. [7] The MOE Key Laboratory of Weak-Light Nonlinear Photonics and TEDA Applied Physics Institute and School of Physics, Nankai University, Tianjin 300457, China. Yujia Liang and Yong Xie contributed equally to this work. Correspondence and requests for materials should be addressed to K.D. (email: kdeng@nanoctr.cn) or to X.W. (email: wuxc@nanoctr.cn) or to I.I.S. (email: Ivan.Smalyukh@colorado.edu) or to Q.L. (email: liuq@nanoctr.cn)

Self-assembly is a ubiquitous phenomenon in nature, creating the diversity of materials and organisms. However, the functionalities and structural complexity of artificial self-assembly are dwarfed by the counterparts in nature, especially in the nanoscale. With rapid development of nanotechnologies, nanoparticle self-assembly exhibits great potentials in bio-assays, optoelectronic devices, sensors, solar cells and so forth[1–7], and thus stimulates great efforts on this field in the past two decades. In nano-assemblies, there exist four key synthetic challenges so far: achieving low defect density, large scale, process controllability, and architecture designability. Up to now, great progresses have been achieved in the first three aspects with the advancements in assembly technologies[8–12], while the last one remains a significant challenge.

As the most important common concern of architecture designability, symmetry control is of importance as it diversifies superstructures and enriches the functional applications[13–16]. In general, shape complementarity determines the final assembly symmetry based on the balance of weak colloidal forces (electrostatic, van der Waal, depletion, steric, interfacial force, adhesion, capillary, and other forces)[8, 17–20]. For example, cubic building blocks lead to tetragonal superlattices, and spherical ones form close-packed hexagonal superlattices[8, 10, 12, 21]. Such symmetry realization is limited by its own intrinsic shape; accordingly, the applications are confined. Due to the complex collective effects of surfaces and the thermodynamically-determined minimum potential energy[22–24], breaking through the shape-modulated symmetry is a formidable task and seems to be unachievable. It necessitates the exploration of new ways to realize the shape-independent symmetry control. If we could introduce a strong force to dominate the assembly process, it might be possible to achieve a desired configuration with the minimum potential energy, and thus control the superlattice symmetry.

The assemblies of noble metal nanorods, especially gold nanorods (GNRs), have exhibited unique anisotropy-dependent optical and optoelectronic properties and accordingly attracted much of the recent interest[25–31]. Owing to such anisotropy, modulating the symmetry of the metal nanorod assembly is more challenging than for the shape-isotropic nanoparticles[32, 33]. In case of GNRs, albeit great advances, only hexagonal symmetry structures[27, 29, 34, 35] have been obtained, and other symmetry structures have yet to be achieved.

Here we present an assembly strategy to achieve symmetry control of the GNRs superlattice. By introducing a governing force, we realize an unexpected tetragonal symmetry of GNR assembly. Such a tetragonal superlattice exhibits unusual thermostability, which is much higher than its hexagonal counterpart. Multiscale modeling reveals that the governing force dominates the interactions involved in the assembly process and determines the thermodynamically favored symmetry of superstructures, i. e., the tetragonal superlattice herein. Such self-assembly strategy has been successfully extended to fabricate the tetragonal superlattices of Ag and Pd nanorods, indicating that the strategy is effective and broadly applicable for building, designing and controlling the colloidal superstructures.

## Results

**Symmetry transformation from hexagonal- to tetragonal-superlattice**. Till now, only hexagonal symmetry of GNR superlattice has been achieved, which is determined by the principle of minimum potential energy. The as-prepared GNRs suspension was centrifuged once and then ultrasonically dispersed in water as cetyltrimethylammonium bromide (CTAB) fully-covered GNRs (FC GNRs, Methods). Using FC GNRs, a typical hexagonal GNR assembly obtained in quasi-equilibrium condition (Fig. 1a) is demonstrated, which is in agreement with the previous

results[27, 29, 34, 35]. The measured unit cell parameters are $a = b = 20.8 \pm 1.4$ nm with $\alpha = 60° \pm 3°$ (Supplementary Note 1, Supplementary Figs. 1 and 2). The as-prepared GNRs suspension was centrifuged twice and then ultrasonically dispersed in water as CTAB partially-covered GNRs (PC GNRs, Methods). In order to modulate the assembly symmetry, Rhodamine 6G (R6G) is chosen to modify PC GNRs. By introducing R6G, the hexagonal GNR superlattice is successfully transformed to a tetragonal GNR superlattice ($a = b = 21.4 \pm 1.3$ nm and $\alpha = 90° \pm 3°$, Fig. 1b). High resolution TEM (HRTEM) images of PC GNRs indicate side facets are composed by two sets of {100} and {110} (Supplementary Fig. 3), which is consistent with the conventional morphological model[36, 37].

To understand the role of R6G in modulation of the assembly symmetry, we investigate the interaction between R6G and GNR. For FC GNRs, the extinction spectra display no observable change in the longitudinal surface plasmon resonance (LSPR) band upon sequential adding of R6G, suggesting adsorption of R6G on rod surface is negligible (Fig. 1c). For PC GNRs, the scenario is quite different (Fig. 1d). With increasing R6G concentration, the LSPR band is gradually red-shifted with a slight decrease in intensity because the adsorption of R6G increases the local dielectric constant on rod surface. Such difference between FC GNRs and PC GNRs is further supported by Raman spectra (Fig. 1e). For FC GNRs, no R6G vibration peaks appear except the Au-Br vibration peak at 177 cm$^{-1}$ originating from the coated CTAB bilayer[38]. For PC GNRs, strong characteristic peaks of R6G at 612, 1364 and 1510 cm$^{-1}$ are observed, indicating that the adsorption of R6G molecules on rod surface and their standing upright thereby[39]. The Au-Br vibration peak at 177 cm$^{-1}$ is also visible, confirming the co-existence of CTAB and R6G on rod surface. Zeta potential of FC GNRs is $45.2 \pm 0.9$ mV and shows no detectable variation with different R6G concentrations (Fig. 1f), substantiating the ignorable interaction between R6G and FC GNR. For PC GNRs, Zeta potential reduces to $28.9 \pm 0.5$ mV, suggesting that CTAB bilayer on rod surface is partially removed after centrifugation. Interestingly, with increasing R6G concentration, we have not observed the influence of adsorbed R6G on the Zeta potential of the rods (Fig. 1f). This is because the Zeta potential of the rods is mainly determined by the CTAB bilayer due to the following two reasons: the thickness of CTAB bilayer is much larger than the length of the single R6G molecule, and the coverage of CTAB on PC GNRs is still dominated. This means that R6G molecules are only adsorbed on the vacant sites where CTAB bilayers have been removed from the rod. Thus, the influence of R6G on Zeta potential can be ignored. Overall, R6G could be adsorbed on PC GNRs (Fig. 1g), and such adsorption is the basis for achieving the tetragonal superlattice.

**Enhanced thermostability of tetragonal superlattice**. Unexpectedly, the tetragonal superlattice exhibits higher thermostability than hexagonal counterpart. For the hexagonal superlattice obtained at room temperature (25 °C, Fig. 2a), after 150 °C 4 h annealing, the original symmetry remains except for the appearance of a few fused particles (Fig. 2b). However, at the higher annealing temperature of 210 °C (4 h), the hexagonal superlattice is totally damaged and melted to the irregular large particles due to the fusion of GNRs (Fig. 2c). For the tetragonal superlattice obtained at 25 °C (Fig. 2d), the symmetry also remains at 150 °C annealing (4 h, Fig. 2e). Strikingly, distinctive difference for the tetragonal superlattice occurs after 210 °C annealing for 4 h. In comparison with the hexagonal counterpart, the tetragonal superlattice survives (Fig. 2f), indicating it possesses a higher thermostability. Furthermore, in order to get more

insights into the thermostability of the tetragonal superlattice, we have performed in situ observations with annealing temperature from 260–310 °C (Fig. 2g). In this temperature range, the tetragonal symmetry remains. However, with increasing temperature, the structural integrity of tetragonal superlattices is gradually destroyed. To confirm that the adsorbed R6G is responsible for the enhancement of thermostability of tetragonal superlattice, we have performed its Raman spectra. Owing to the plasmonic

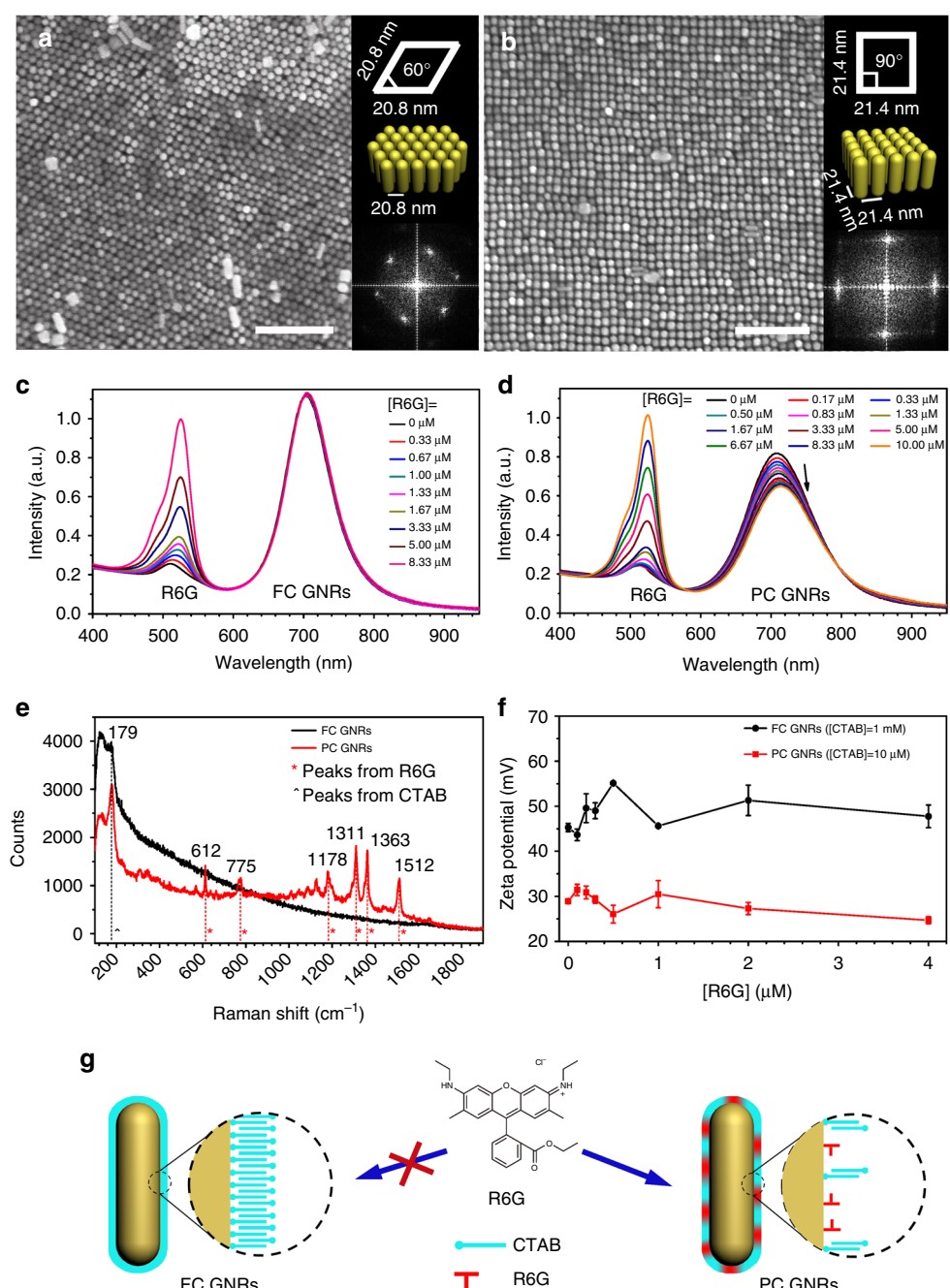

**Fig. 1** R6G-mediated assembly of a tetragonal superlattice of GNRs. **a** A typical SEM image of hexagonal superlattice (scale bar, 200 nm). Inset (top) presents the corresponding unit cell with parameters ($a = b = 20.8$ nm, $\alpha = 60°$). Inset (middle) shows 3D schematics of the hexagonal superlattice. Inset (bottom) is the corresponding Fast Fourier Transformation (FFT) image, confirming hexagonal symmetry of the superlattice. **b** A typical SEM image of tetragonal superlattice (scale bar, 200 nm). Inset (top) presents the corresponding unit cell with parameters ($a = b = 21.4$ nm, $\alpha = 90°$). Inset (middle) shows 3D schematics of the tetragonal superlattice. Inset (bottom) is the corresponding FFT image, confirming tetragonal symmetry of the superlattice. **c** Effects of R6G on extinction spectra of FC GNRs aqueous suspension. Upon sequential adding of R6G, LSPR band of FC GNRs displays no observable change. **d** Effects of R6G on extinction spectra of PC GNRs aqueous suspension. With increasing R6G concentration, the LSPR band is gradually red-shifted with a slight decrease in intensity. **e** SERS spectra of FC and PC GNRs in aqueous solution. Strong characteristic peaks of R6G are observed from PC GNRs, but not from FC GNRs. **f** Zeta potential of FC and PC GNRs in aqueous solution vs. R6G concentration. Reduced Zeta potential of PC GNRs supports the partial coverage of CTAB bilayer on PC GNRs. For both FC and PC GNRs, increasing R6G concentration, no detected influence of R6G on the Zeta potential of the rods is observed. The mean values and error bars are statistically summarized from three measurements. The error bars represent the standard deviations. **g** Schematics of R6G and CTAB on rod surfaces. For FC GNRs, fully-covered CTAB hinders R6G to adsorb on the rod surface. For PC GNRs, CTAB partially-covered rod affords enough space for R6G to adsorb

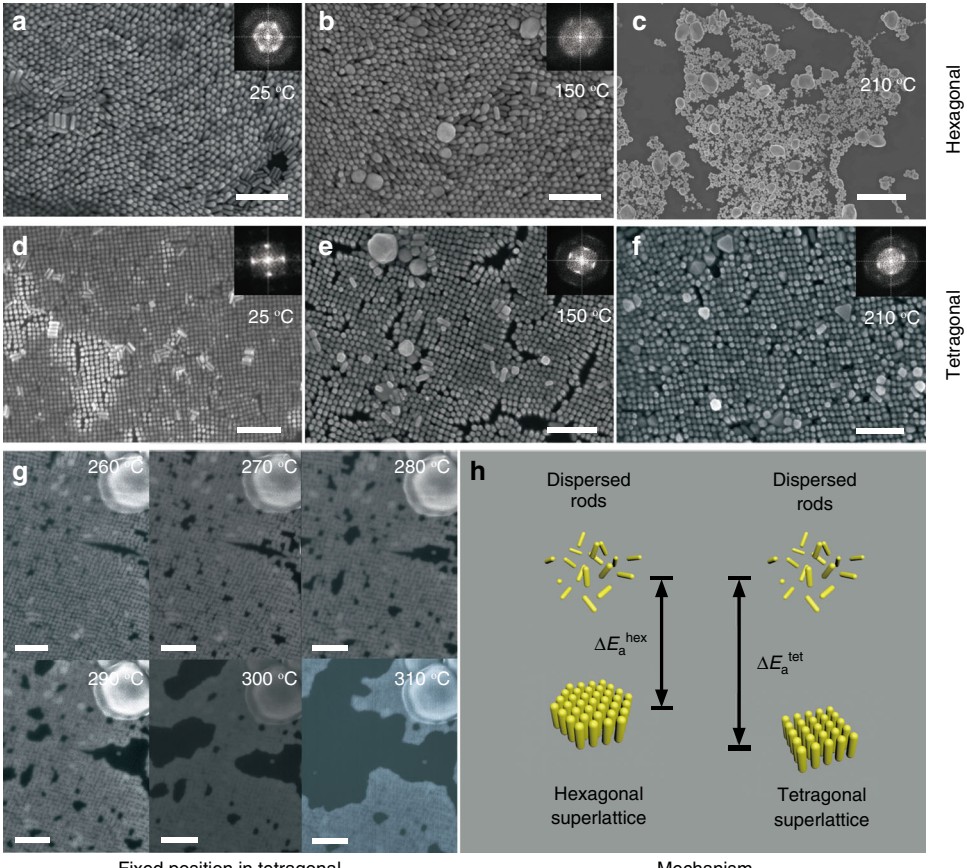

**Fig. 2** Enhanced thermostability of tetragonal GNR superlattice. **a–c** SEM images of hexagonal superlattices after 25, 150 and 210 °C thermal annealing. At two lower temperatures, the hexagonal symmetry remains. At 210 °C, the hexagonal superlattice is totally damaged. The corresponding FFT images in insets show the symmetry change. **d–f** SEM images of tetragonal superlattices after 25, 150 and 210 °C thermal annealing. Till 210 °C, the tetragonal symmetry still survives. The corresponding FFT images in insets indicate that tetragonal symmetry remains. **g** SEM images of the tetragonal superlattice at a fixed position from 260 to 310 °C, showing the tetragonal superlattice is stable until 310 °C. **h** Schematic energy states of hexagonal and tetragonal superlattices, suggesting more energy is required for destroying tetragonal superlattice than hexagonal one. Scale bar in **c** is 2 μm and 200 nm for others

hotspots in the tetragonal superlattice, much stronger R6G Raman signals in tetragonal superlattice are detected (Supplementary Fig. 4) than those of the discrete GNRs dispersed in the aqueous solution (Fig. 1e), suggesting the confinement of R6G between the adjacent nanorods. Furthermore, such Raman signals even can be observed after 250 °C thermal annealing, indicating the critical role of R6G in maintaining tetragonal superlattice. Based on the above observations, it is reasonable to conclude that more energy is required for destroying tetragonal superlattice than hexagonal one (Fig. 2h).

**Multiscale modeling and origin of governing force**. In our system, hexagonal superlattice assembly involves electrostatic, van der Waals and depletion forces (Fig. 3a, b, Supplementary Fig. 5), while in tetragonal superlattice, an additional force originating from R6G should be considered (Fig. 3c). The total potential $U$ for the interactions between two side-by-side metal nanorods can be expressed as

$$U = U_{ele} + U_{van} + U_{dep} + U_{chem}, \quad (1)$$

where $U_{ele}$, $U_{van}$, $U_{dep}$ and $U_{chem}$ denote the electrostatic potential, the van der Waals potential, the depletion potential, and the potential originated from R6G interaction, respectively. For the potentials ($U_{ele}$, $U_{van}$, $U_{dep}$) between GNRs, we adopt the conventional descriptions (Supplementary Notes 2–4). Our

calculations indicate that for the side-by-side GNR dimer without R6G, an energy minimum (−1.848 eV) appears at a separation distance ~1.047 nm (Supplementary Fig. 6). Here we use the energy per unit area to describe the thermodynamic stability of the assemblies. The energy per unit area of the hexagonal superlattice is −3.799 meV nm$^{-2}$, which is lower than that of the tetragonal one (−3.290 meV nm$^{-2}$). This explains why only the hexagonal superlattice was obtained previously. However, when R6G is introduced, $U_{chem}$ needs to be incorporated. Based on our density functional theory (DFT) calculations (Supplementary Note 5), R6G could adsorb on the metal nanorods vertically through the Br ions on rod surface (Fig. 3c and Supplementary Figs. 8 and 9). Furthermore, R6G molecules interact with each other in a head-to-tail fashion via π-π stacking and hydrogen bonding (Supplementary Fig. 10), similar to the reported R6G J-aggregates on particle surface. And the interaction is directional and strong (−1.444 eV), resulting in the formation of R6G chains (Fig. 3c and Supplementary Fig. 10). Once R6G chain is formed between two GNRs, a new minimum (−3.155 eV) at the separation of 1.250 nm occurs (Fig. 3d). Obviously, such a force is a governing force and dominates over all the other forces involved in the system.

Since both the adsorbed R6G chain and the governing force are vertical to the rod side surface, the side facet should be considered. TEM measurements were performed for the tetragonal superlattices (Fig. 3e). In a unit cell of tetragonal

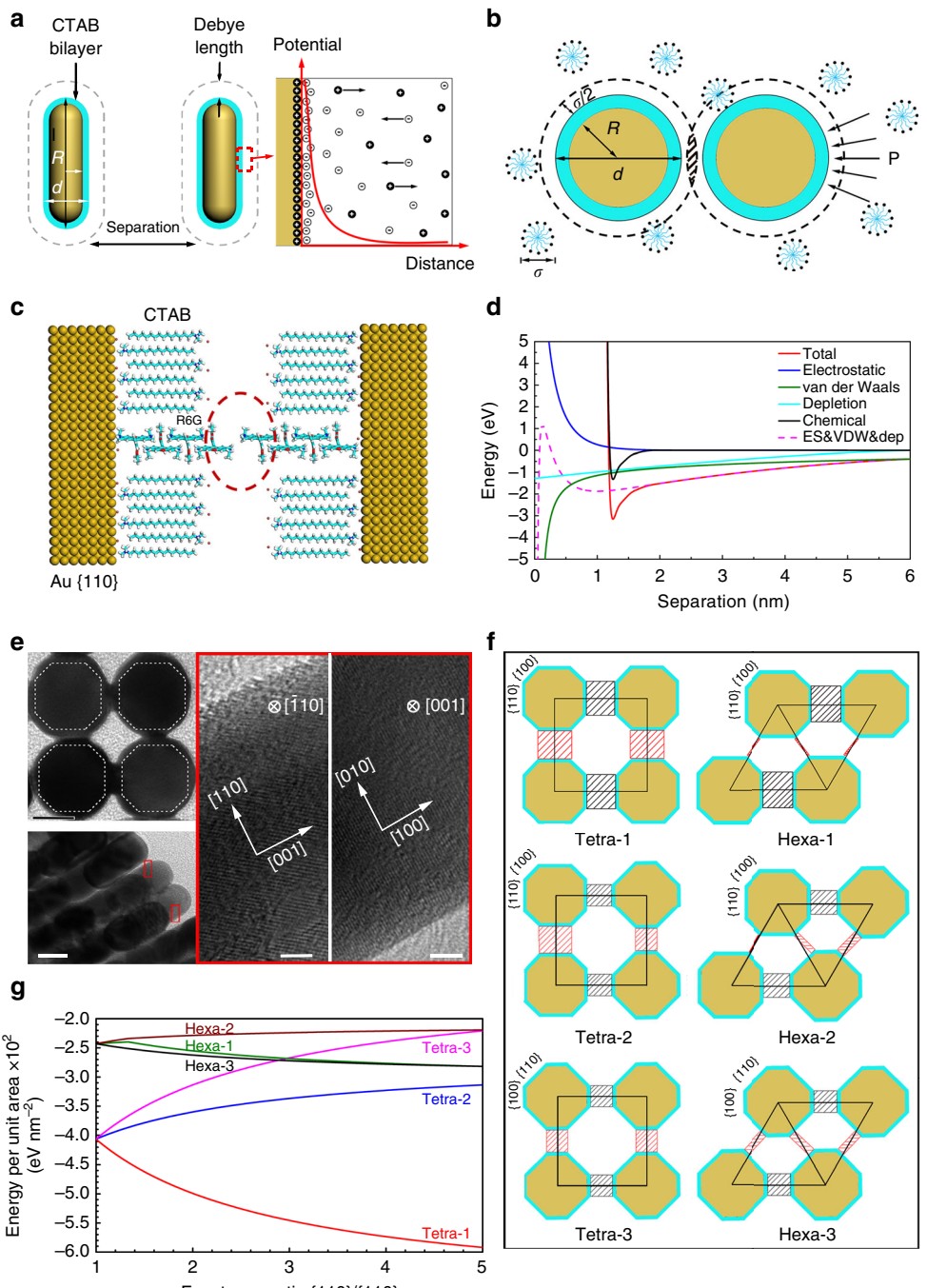

**Fig. 3** Theoretical simulations of interactions involved in superlattices. **a** Electrostatic and van der Waals interactions in GNR dimer. **b** Top view of depletion interaction in GNR dimer. **c** Diagram for R6G chain formed between GNR dimer by density functional theory (DFT) method. **d** Potential energy for GNR dimer with R6G. **e** Upper left: TEM image of a unit cell of tetragonal superlattice (scale bar, 10 nm), lower left: TEM image of side view of the tetragonal superlattice (scale bar, 20 nm), right: HRTEM images of the red box in lower left panel (scale bar, 2 nm). Electron beam is aligned on [−110] direction and [001] direction, respectively. **f** Unit cells of tetragonal and hexagonal superlattices with different dimer combinations. **g** Effects of facet area ratio on energy per unit area for tetragonal and hexagonal superlattices

superlattice, four nanorods are linked face-to-face and the cross-section of nanorod can be described as an octagon. Further, HRTEM images show the side facets are linked by {100} or {110} ones. Based above TEM observations, we consider nanorod is surrounded by four {100} and four {110} side facets. Adsorption energies for R6G on Au{110} and Au{100} facets are similar (Supplementary Table 1), suggesting that adsorption of R6G on these two facets is equivalent. Obviously, the alignment of two

nanorods with larger facets face-to-face can guarantee more R6G chains formed between two nanorods no matter the larger facet is {110} or {100} facet. If {110} facet is larger, two nanorods with opposite {110} facets (denoted as {110}–{110} dimer) would be energetically more favorable than {110}–{100} dimer and {100}–{100} dimer. For the most stable {110}–{110} dimer, the pivotal step to form the hexagonal superlattice or tetragonal superlattice is determined by the location of the next {110}–{110}

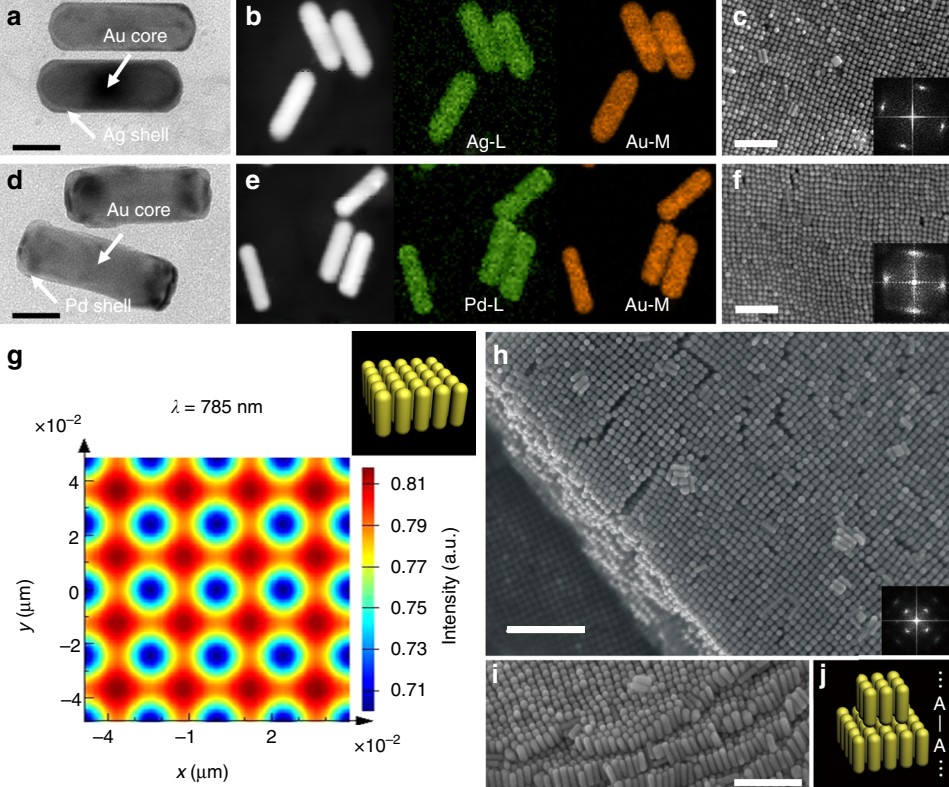

**Fig. 4** R6G-induced Ag and Pd tetragonal superlattices and plasmonic enhancement. **a** TEM characterization of $Ag_{shell}Au_{core}$ nanorods (scale bar, 20 nm). **b** High-angle annular dark field (HAADF)-STEM image and element mapping of $Ag_{shell}Au_{core}$ nanorods. **c** SEM images of $Ag_{shell}Au_{core}$ nanorods tetragonal superlattice (scale bar, 160 nm). **d** TEM characterization of $Pd_{shell}Au_{core}$ nanorods (scale bar, 20 nm). **e** HAADF-STEM image and element mapping of $Pd_{shell}Au_{core}$ nanorods. **f** SEM images of $Pd_{shell}Au_{core}$ nanorods tetragonal superlattice (scale bar, 150 nm). **g** The simulated electrical field intensity enhancement of GNR tetragonal superlattice at 785 nm. The inset shows 3D schematic of single layer of tetragonal superlattices. **h** SEM image of GNR multilayer superlattice (top view, scale bar, 200 nm). **i** SEM image of GNR multilayer superlattice (side view, scale bar, 150 nm). **j** Schematics of GNR tetragonal superlattices with head-to-head alignment between neighboring layers

dimer. The probability of forming different superlattices can be correlated to the overlap area between two neighboring facets in tetragonal and hexagonal superlattices with two {110}–{110} dimers (Tetra-1 and Hexa-1, Fig. 3f). The overlap area (red shadow, Fig. 3f) between two {110}–{110} dimers in Hexa-1 is very small, showing a much lower probability to form the linkage of R6G chain than that in Tetra-1. The other facets combinations also support much lower probabilities of forming the hexagonal superlattices. Energies per unit area of different superlattices confirm that the Tetra-1 is the most stable formation, and with increasing of {110}/{100} ratio the energy per unit area of Tetra-1 decreases accordingly (Fig. 3g). Especially, when the area of {110} and {100} is the same, the formation probability of the tetragonal supperlattice will reach the maximum. Therefore, the tetragonal superlattice is the most thermodynamically favorable, strongly supporting our experimental findings.

It should be noted that choosing the introduced molecules is of critical importance. When replacing R6G with Rhodamine B (RhB), only random rod aggregates are observed (Supplementary Fig. 11). This is because RhB prefers to form side-by-side dimers via hydrogen bonding rather than rod-like chains in head-to-tail fashion (Supplementary Fig. 10), which is unable to provide a directional linkage between two rods.

**Extension of assembly strategy to Ag and Pd superlattices**. On the basis of the above theoretical analyses, modulation of super-lattice symmetry guided by the governing force should be a phenomenon of general significance. In principle, changing GNRs to other metal rods, such symmetry control should be able to be realized. Based on our calculation, for Ag and Pd nanorods, introducing R6G can also result in the similar potential curves between the rod dimer (Supplementary Fig. 7). In order to confirm the above theoretical results, we performed assembly of Ag and Pd nanorods experimentally. To ensure the similar morphology of the nanorods, we employed GNRs as the template to guide the epitaxial growth of Ag and Pd. The morphology and element mapping of $Ag_{shell}Au_{core}$ nanorods have been characterized by TEM (Fig. 4a, b). As expected, after introducing R6G, the $Ag_{shell}Au_{core}$ nanorods also form the tetragonal superlattice (Fig. 4c). Similarly, $Pd_{shell}Au_{core}$ nanorods (Fig. 4d, e) also form the tetragonal superlattice (Fig. 4f). This suggests that the strategy is applicable for the nanoparticles made of other noble metals.

Different superlattice symmetries may lead to diverse applications. As an example, we employ Finite-Difference Time-Domain (FDTD) to simulate field enhancement of the nanorod super-lattices (Supplementary Note 6), and demonstrate the electromagnetic field enhancements of tetragonal and hexagonal superlattices for Au, Ag, and Pd nanorods at three laser wavelengths often-used in Raman spectrophotometer (Fig. 4g, Supplementary Figs. 12–14). Both symmetries show field enhancement in a broad spectrum due to strong plasmonic coupling. Although the field enhancement for monolayer in tetragonal superlattice is slightly stronger than that in hexagonal one, the enhancement will increase significantly and more

plasmon modes will occur in the multilayer of the tetragonal superlattice. Experimentally, by tuning assembly conditions, we also obtain the multilayer of GNR tetragonal superlattice (Fig. 4h, i), in which the GNRs between the neighboring layers align head-to-head (Fig. 4j). The unique head-to-head alignment between neighboring layers affords an experimental solution for a simulated novel nanolens, which is made of stacked silver nanorods (50 nm-long nanorods head-to-head alignment with a gap of 10 nm) and applied to colour far-field observation of the individual viruses and other nano-entities[7].

In summary, we present a strategy to tune the assembly symmetry of nanorods by introducing a governing force. With the help of R6G molecules, we realize tetragonal superlattice of the nanorods, which breaks through the only hexagonal symmetry of the superlattice so far. This unexpected tetragonal superlattice exhibits an enhanced thermostability as compared to its hexagonal counterpart. By combining DFT and conventional modeling of colloidal interactions, we reveal that the governing force arising from R6G dominates the interactions involved in the assembly process, which results in a new potential minimum. Vertically adsorbed R6G molecules demand the nanorods to be arranged face-to-face in order to maximize the R6G interaction, leading to the most stable non close-packed tetragonal super-lattice, in agreement with the observation of its enhanced thermal stability. Both theoretical and experimental results suggest that our strategy is widely applicable and may open up a new avenue for realization of diverse assembly structures with pre-designed properties. This will be instrumental for designing, realizing and controlling the assembly of composite molecular-colloidal nanomaterials and may greatly expand their future applications.

## Methods

**Chemicals**. Rhodamine 6G (R6G) was purchased from J&K Chemical. Cetyl-trimethylammonium bromide (CTAB) was purchased from Amresco. All the other chemicals used in the experiments were purchased from Sigma-Aldrich without further purification.

**Preparation of the GNRs**. CTAB-coated GNRs were synthesized by a well-developed seed-mediated growth method[40, 41]. The obtained GNRs (average aspect ratio = 3.5, length ≈60 nm, diameter ≈17 nm) were centrifuged (9000 rpm for 7 min, 25 °C) and the supernatant were removed. By keeping the same cen-trifugation conditions (such as centrifugation force, centrifugation time, cen-trifugation tube), uniform CTAB desorption from the rods is guaranteed. The precipitate was then re-dissolved in deionized (DI) water (18 MΩ·cm) to a con-centration of ~ 0.5 nM for GNRs and 1.0 mM for CTAB. Such GNRs suspension is called CTAB fully-covered GNRs (FC GNRs) herein. The as-prepared GNRs sus-pension was centrifuged for the second time (12,000 rpm for 5 min, 25 °C). The 10 μL precipitate was ultrasonically dispersed and is termed as CTAB partially-covered GNRs (PC GNRs) with a concentration of ~0.5 nM for GNRs and 1.0 μM for CTAB. The concentration of the GNRs was determined by Beer-Lambert law, $A = \varepsilon c l$, with a molar extinction coefficient of $\varepsilon = 4.6 \times 10^9 \, M^{-1} \, cm^{-1}$ at 707 nm[42].

**Preparation of the $Ag_{shell}Au_{core}$ NRs**. 1 ml purified GNRs was first mixed with 1 ml CTAB (0.1 M) and 1 ml $H_2O$. Then, $AgNO_3$ (15 μl, 10 mM), NaOH (300 μl, 0.2 mM), and AA (15 μl, 0.1 M) were added to initiate Ag shell growth. The growth was conducted at 30 °C for 12 h. After that, the obtained $Ag_{shell}Au_{core}$ NRs were purified by centrifugation (12,000 rpm for 5 min) twice. The 10 μL precipitate was ultrasonically dispersed and is termed as "CTAB partially covered" $Ag_{shell}Au_{core}$ NRs (PC $Ag_{shell}Au_{core}$) with a concentration of ~ 0.5 nM for rods and 1.0 μM for CTAB.

**Preparation of the $Pd_{shell}Au_{core}$ NRs**. 1 ml purified GNRs was first diluted with 2 ml CTAB (0.1 M) aqueous solution. Then, 65 μl $K_2PdCl_4$ (2.5 mM), 15 μl $H_2SO_4$ (1 M), and 15 μl AA (0.1 M) were added to start Pd shell growth. The growth was conducted at 30 °C for 3 h. After that, the obtained $Pd_{shell}Au_{core}$ NRs were purified by centrifugation (12,000 rpm for 5 min) twice. The 10 μL precipitate was ultra-sonically dispersed and is termed as "CTAB partially covered" $Pd_{shell}Au_{core}$ NRs (PC $Pd_{shell}Au_{core}$) with a concentration of ~0.5 nM for rods and 1.0 μM for CTAB.

**Preparation of tetragonal superlattice**. The assemblies of various NRs were obtained by the controllable droplet drying method[43, 44]. 5 μL 0.5–1.5 mM R6G solutions were first mixed with PC NRs. Then, 10 μL droplet was put on a silicon

wafer which had been cleaned by acetone, ethanol and DI water in advance. A quasi-equilibrium assembly was achieved by placing the sample in a programming climate chamber at 25 °C and changing the humidity from 50–85% gradually. R6G adsorption on rod surface is verified by extinction spectra and SERS. Similar adsorption behaviors on other particle surfaces have been observed[45–49].

**Characterizations**. Scanning electron microscope (SEM, Hitachi S4800) was employed to characterize the nanorod superlattices. A confocal Raman spectro-scope (Renishaw inVia) was employed with an objective lens (50×) at the excitation wavelength of 785 nm. Extinction spectra were recorded on a UV-Vis-NIR spec-trophotometer (Cary50) at room temperature. Zeta potential data were collected using Zetasizer Nano ZS. TEM measurements were taken on Tecnai-F30 with an accelerating voltage of 200 kV.

**Annealing experiments**. The GNR superlattice samples were placed in a tube furnace, which was heated at targeted temperatures for 4 h and cooled down to room temperature, and then observed by SEM. From 260–310 °C, the structural evolution of the nanorod superlattice was monitored at the fixed position by SEM.

**Data availability**. All relevant data are available from the authors on request.

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

## Acknowledgements

This work is partly supported by the National Key Research and Development Program of China (2016YFA0200403 and 2016YFA0200903), NSFC (10974037, 61505038, 51502011, U1613204), the CAS Strategy Pilot Program (XDA 09020300) and Eu-FP7 Project (No. 247644); I.S.S. acknowledges the support of the US Department of Energy award ER46921. C.G. acknowledges the "Guangdong Innovative and Entrepreneurial Research Team Program" under contract No. 2016ZT06G587.

## Author contributions

Q.L., X.W., K.D. and I.I.S. guided this project, analyzed the results and drafted the manuscript. Y.L. and Y.X. performed the experiments, analyzed the results, and drafted the manuscript; K.D. conducted the DFT calculation. D.C. contributed to the FDTD simulation. S.H. and T.W. synthesized the GNRs. C.G. and F.Y. participated partially in performing the experiments.

## Additional information

**Competing interests:** The authors declare no competing financial interests.

