## [Peer Review File · Nature Communications]

Reviewers' comments:

Reviewer #1 (Remarks to the Author):

The manuscript, entitled "Symmetry control of nanoparticle superlattice driven by a governing force," describes an approach for assembling gold nanorods into a tetragonal superlattice. The authors have also evaluated their assembly strategy with Ag or Pd shell core gold nanorods with various simulated data. Although it's well known that the self-assembly of gold nanorods is greatly influenced by shape, drying process, stabilizer surfactants and its relative concentration with respect to nanorods; yet, the results presented by authors are interesting particularly with respect to tetragonal lattice formation. However, I have following apprehensions:

Major comments:

After viewing from the top, the terminal end shape of the gold nanorods in Fig. 1a looks quite different from the Fig 1b. In Fig 1a, most of the nanorods terminal end look like penta shaped whereas in Fig1b is spherical. Had the experiments been done from different batches of prepared gold-nanorods? If this is the case then the results presented for symmetry transformation from hexagonal to tetragonal superlattice will not be comparable.

The process of making R6G adsorbed gold nanorods follows centrifugation of CTAB capped gold nanorods (12,000 rpm for 5 min), removal of CTAB and adsorption of R6G. In this process, it is quite possible that the CTAB will be removed from the gold nanorod surface from random sites and R6G will take over that place. If the positioning of R6G is random on the surface of nanorods; then maintaining uniformity of microenvironment around each nanorod will be very difficult. Without such uniformity, precise control over the self-assembly into particular geometry will be hard to achieve.

Page 5, line 104-106: Authors need to explain in more details about the statement "increasing concentration of R6G, no influence on Zeta potential is observed (Fig. 1f) because the adsorbed R6G may be screened by CTAB bilayer"

Page 5, line 98-100: "The LSPR band in the extinction spectra is gradually red-shifted with its intensity damped when R6G concentration increases, implying the change in the surrounding dielectric environment (Fig.1d)". – This statement needs to be rewritten. Change in the surrounding dielectric environment could also lead to increase in intensity and blue shift.

Page 7, Fig 2a: On the top left and bottom left some tetragonal arrangements of the gold nanorods can be seen which has been synthesized using CTAB gold nanorods only. Why?

Minor comments:

- Description and discussion of figures in main text should be in the sequence as it appears in the figure.
- The manuscript needs comprehensive editing. At many places, the sentences are too long and convoluted. The sentences need to be rewritten in a better way. For example:

Page 2, line 22-24: Self-assembly of nanoparticles..... but control of the symmetry of the ensuing mesoscopic colloidalchallenge.

Page 2, line 24-28: By introducing.....symmetry obtained so far.

Page 2, line 30-33: Multiscale modelling.....its hexagonal counterpart.

Page 3, line 43-46: With rapid.... information storage, and so forth.

- Page 4, line 74-77: The following statement should be rephrased: "Such self-assembly strategy has been successfully extended to a..... universal and broadly applicable for...superstructures". – To be universal the process should be applicable to all systems including the surfactant. However, based on the results the current process did not work when R6B was replaced by RB.

Reviewer #2 (Remarks to the Author):

The manuscript 'Symmetry control of nanoparticle superlattice driven by a governing force' presents experimental data on the self-assembly of gold nano-rods (GNR) into tetragonal structures, which display superior thermal stability to similar assemblies with hexagonal arrangements. The findings are supported by simulation studies.

In detail: The authors show that DNRs that are stabilised via a double-layer of the common surfactant CTAB will spontaneously self-assemble into a layered structure with each layer having a hexagonal symmetry, as would be expected for hard-rod behaviour forming a smectic liquid crystal at high enough concentrations. Their surprising finding is, that when the sterically stabilising layer of CTAB is partially replaced by the dye molecule Rhodamine such that it intercalates into the CTAB double-layer the effective inter-particle interactions change. As a result the particles will now self-assemble into a tetragonal symmetry, meaning the smectic layers have a square symmetry now. This is reminiscent of the self assembly of micelles made of Pluronics (PEO-PPO-PEO), which show the less dense BCC packing for shorter chains with low aggregation number, while longer chains with higher aggregation number will display FCC packing, because they behave more like hard spheres. The author's simulation studies on the effective interaction potential support the idea that a less dense packing will be assumed for particles with a 'softer' potential. In analogy, Landau and Lifshitz have argued that atoms with a more soft potential will form BCC (for spherical symmetry).

In addition the authors also show that the tetragonally assembled GNR-crystals display superior stability against melting. This may again be due to the fact that the Rhodamine dye remains stabilising the rods as opposed to the purely CTAB-stabilized particles, which start to melt and fuse at lower temperatures.

Both results are new and very exciting, and therefore deserve publication.

However, I do ask the authors to have the English improved considerably.

Point to point response letter

Reviewer #1:

Q1: After viewing from the top, the terminal end shape of the gold nanorods in Fig. 1a looks quite different from the Fig 1b. In Fig 1a, most of the nanorods terminal end look like penta shaped whereas in Fig1b is spherical. Had the experiments been done from different batches of prepared gold-nanorods? If this is the case then the results presented for symmetry transformation from hexagonal to tetragonal superlattice will not be comparable.

A1: Thanks for the careful observation on the graphs. GNRs as shown in Supplementary Figure 3 (HRTEM images of PC GNRs) are single crystal, determining their symmetry is impossible to be five-fold symmetry. The different terminal end shapes of the gold nanorods in Fig.1a and Fig. 1b are caused by the differences in image conditions, such as the imaging angle and the image magnification. In addition, we have checked our experimental records again and further confirmed that the same batch of rods was used in Fig.1a and Fig. 1b. We thank the referee for this comment.

Q2: The process of making R6G adsorbed gold nanorods follows centrifugation of CTAB capped gold nanorods (12,000 rpm for 5 min), removal of CTAB and adsorption of R6G. In this process, it is quite possible that the CTAB will be removed from the gold nanorod surface from random sites and R6G will take over that place. If the positioning of R6G is random on the surface of nanorods; then maintaining uniformity of microenvironment around each nanorod will be very difficult. Without such uniformity, precise control over the self-assembly into particular geometry will be hard to achieve.

A2: We appreciate the remarks. As shown in **Preparation of the GNRs of Methods**, the “CTAB partially covered” GNRs were obtained *via* centrifugation process. In such process, with the same centrifugation condition (such as centrifugation force, centrifugation time, centrifugation tube), we think CTAB desorption from the rods is uniform rather than random. We have made the revision in the main text and **Methods** to clarify this further.

Q3: Page 5, line 104-106: Authors need to explain in more details about the statement “increasing concentration of R6G, no influence on Zeta potential is observed (Fig. 1f) because the adsorbed R6G may be screened by CTAB bilayer”

A3: We are grateful for this helpful remark. For the “CTAB partially covered” GNRs, with increasing concentration of R6G, we have not observed the influence of adsorbed R6G on the Zeta potential of the rods (Fig. 1f). This is because the Zeta potential of the rods is mainly determined by the CTAB bilayer due to the following two reasons. First, the thickness of CTAB bilayer is much larger than the length of the single R6G molecule. Second, the coverage of CTAB on the rods is still dominated. Thus, the influence of R6G on Zeta potential can be ignored. We have explained it in more details in the revised manuscript.

Q4: Page 5, line 98-100: “The LSPR band in the extinction spectra is gradually red-shifted with

its intensity damped when R6G concentration increases, implying the change in the surrounding dielectric environment (Fig.1d)”. – This statement needs to be rewritten. Change in the surrounding dielectric environment could also lead to increase in intensity and blue shift.

A4: Following this suggestion, we have rewritten the sentence to “With increasing R6G concentration, the LSPR band is gradually red-shifted with a slight decrease in intensity (Fig.1d) because the adsorption of R6G increases the local dielectric constant on rod surface.”.

Q5: Page 7, Fig 2a: *On the top left and bottom left some tetragonal arrangements of the gold nanorods can be seen which has been synthesized using CTAB gold nanorods only. Why?*

A5: We have zoomed in the top left and bottom left of Fig. 2a as shown below and carefully checked the parts. Herein the superlattice of nanorods is defined well using the hexagonal unit cell (the yellow rhombus). We believe some seeming tetragonal arrangements of the gold nanorods are due to the difference of viewing angle.

Q6: - *Description and discussion of figures in main text should be in the sequence as it appears in the figure.*

A6: We have revised the description and discussion of the figures in the sequences, which appear in Fig. 1, Fig. 2, and Fig. 4.

Q7: - *The manuscript needs comprehensive editing. At many places, the sentences are too long and convoluted. The sentences need to be rewritten in a better way. For example:*

Page 2, line 22-24: Self-assembly of nanoparticles..... but control of the symmetry of the ensuing mesoscopic colloidalchallenge.

Page 2, line 24-28: By introducing.....symmetry obtained so far.

Page 2, line 30-33: Multiscale modelling.....its hexagonal counterpart.

Page 3, line 43-46: With rapid.... information storage, and so forth.

A7: We have carefully revised the manuscript overall, especially focusing on the above sentences.

Q8: - Page 4, line 74-77: The following statement should be rephrased: “Such self-assembly strategy has been successfully extended to a..... universal and broadly applicable for...superstructures”. – To be universal the process should be applicable to all systems including the surfactant. However, based on the results the current process did not work when R6B was replaced by RB.

A8: Thanks for your suggestion. We have removed “universal” in the full text.

Reviewer #2:

Q1: However, I do ask the authors to have the English improved considerably.

A1: Thanks for your helpful suggestion. We have carefully revised the manuscript to improve English and clarity.

REVIEWERS' COMMENTS:

Reviewer #1 (Remarks to the Author):

The authors properly addressed the reviewer's comments and concerns; and the reviewer recommends accepting the manuscript.